# Association between albumin-corrected anion gap and in-hospital mortality of intensive care patients with trauma: A retrospective study based on MIMIC-III and IV databases

**Fei Yin**[1], **Zhenguo Qiao**[2], **Xiaofei Wu**[1], **Qiang Shi**[1], **Rongfei Jin**[1], **Yuzhou Xu**[3]*

**1** Department of Emergency, Suzhou Ninth People's Hospital, Suzhou Ninth Hospital Affiliated to Soochow University, Suzhou, Jiangsu, China, **2** Department of Gastroenterology, Suzhou Ninth People's Hospital, Suzhou Ninth Hospital Affiliated to Soochow University, Suzhou, Jiangsu, China, **3** Department of Orthopedics, Suzhou Ninth People's Hospital, Suzhou Ninth Hospital Affiliated to Soochow University, Suzhou, Jiangsu, China

* yuzhou_xu0303@163.com

## Abstract

### Background

To investigate the correlation between albumin-corrected anion gap(ACAG) within the first 24 hours of admission and in-hospital mortality in trauma patients in intensive care unit (ICU).

### Materials and methods

We utilized the MIMIC-III and MIMIC-IV databases to examine trauma patients admitted to the ICU. The relationship between ACAG and in-hospital mortality in trauma patients was analyzed using Receiver Operating Characteristic(ROC) curve, Kaplan-Meier (K-M) survival curve, and Cox regression model. Propensity score matching (PSM) and subgroup analysis were conducted to enhance stability and reliability of the findings. Mortality at 30-day and 90-day served as secondary outcomes.

### Results

The study enrolled a total of 1038 patients. The AUC for ACAG (0.701, 95%CI: 0.652–0.749) was notably higher than that for anion gap and albumin. The Log-rank test revealed that the optimal cut-off point of ACAG for predicting in-hospital mortality was determined to be 20.375mmol/L. The multivariate Cox regression analysis demonstrated an independent association between high ACAG level and a higher risk of in-hospital mortality (HR = 3.128, 95% CI: 1.615–6.059). After PSM analysis, a matched cohort consisting of 291 subjects was obtained. We found no signifcant interaction in most stratas. Finally, The in-hospital, 30-day, and 90-day survival rates in the high ACAG group exhibited a statistically decrease compared to those in the low ACAG group both pre- and post-matching.

**Data Availability Statement:** All relevant data are within the manuscript and its Supporting Information files.

**Funding:** The author(s) received no specific funding for this work.

**Competing interests:** The authors have declared that no competing interests exist.

## Conclusion

The elevated level of ACAG was found to be independently associated with increased in-hospital mortality among trauma patients in the ICU.

## Introduction

Trauma ranks as the fourth leading cause of death worldwide, accounting for approximately 10% of all mortalities [1]. More than 2.8 million individuals were annually hospitalized in the United States due to trauma [2]. Hemorrhage caused by trauma often leads to shock, which was a primary contributor to early mortality among trauma patients. During the initial stages of trauma, it could be challenging to detect occult shock based on general clinical manifestations and physiological parameters such as heart rate, blood pressure, respiratory rate, and urine volume. Additionally, individuals with hypertension, atherosclerosis or prolonged use of certain cardiovascular medications may exhibit delayed responses to shock [3, 4]. Tissue hypoxia and hypoperfusion under shock conditions result in severe metabolic acidosis. The combination of acidosis along with hypotension and coagulopathy was referred to as the "trauma triad of death", which significantly predicts adverse outcomes within 24 hours following trauma [5].

The Anion gap (AG) reflected the disparity between unmeasured cations and anions concentration in serum and serves as one of the most commonly utilized biomarkers for diagnosing acid-base imbalances and identifying causes of metabolic acidosis. A study conducted by Ahmed et al. demonstrated that AG was an independent prognostic factor for severe trauma patients with an adjusted hazard ratio (HR) of 2.460 [6]. However, literature had noted that during the first hour after hemorrhagic shock onset, there was a greater increase in anion gap compared to serum lactate, this discrepancy may be attributed to uncorrected serum albumin's influence on AG [7, 8]. Anion Gap Corrected for Albumin (ACAG) represents AG values adjusted according to ALB [9], potentially offering improved evaluation capabilities regarding metabolic acidosis and prognosis among trauma patients. The scarcity of previous studies on this topic necessitated the present study, which aimed to ascertain whether ACAG can provide a more accurate prediction of trauma outcomes.

## Method

### Database

This study was a retrospective analysis, utilizing data from MIMIC-Ⅲ Clinical Database CareVue subset and MIMIC-Ⅳ v2.2 databases. The patient population consisted exclusively of individuals admitted to intensive care units at the Beth Israel Deaconess Medical Center (BIDMC) in Boston, Massachusetts. The MIMIC-Ⅲ Clinical Database CareVue subset was derived from the lager MIMIC-Ⅲ Clinical Database v1.4, encompassing patients admitted between 2001 and 2008 [10]. MIMIC-Ⅳ v2.2 included 299,712 patients from 2008 to 2019 [11]. These two databases were mutually exclusive without any overlap or interference. The study involved an analysis of a de-identified publicly available database, which had received prior approval from the Institutional Review Board (IRB) at MIT and Beth Israel Medical Center. No additional ethics approval was necessary. The CITI Program course on Human Research and Data or Specimens Only Research had been successfully completed by us in order to obtain permission for accessing the databases (Record ID: 41,696,976). All individual

patient information within these databases remained anonymous, with the exemption of ethical review and informed consent.

## Patients

In this study, we utilized Structured Query Language (SQL) to extract data from the MIMIC-III CareVue subset and MIMIC-IV databases using Navicat Premium (version 16.1.11). Patients meeting the criteria of having an initial diagnosis corresponding to trauma diagnosis codes in either ICD-9 (ranging from 800 to 959) or ICD-10 (ranging from S00-S99, T00-T14, and T20-T32) were included [12, 13]. In cases where patients had multiple admissions to the ICU, only the first admission was considered. The screening criteria consisted of: (1) excluding patients aged <18 years or >89 years; (2) excluding patients with an ICU stay duration of less than 24 hours; (3) excluding patients with missing important data such as AG, ALB, ACAG within 24 hours of ICU admission. Recorded variables included age, sex, race, Sequential Organ Failure Assessment (SOFA) score, Acute Physiology Score III (APS III), Glasgow Coma Scale (GCS) score, Simplified Acute Physiology Score II (SAPS II), Oxford Acute Severity of Illness Score (OASIS), comorbidities, and other clinical data. Additionally recorded were hematocrit, hemoglobin, platelet counts, white blood cell counts(Wbc), albumin(ALB), anion gap (AG), albumin corrected anion gap(ACAG), bicarbonate, urea nitrogen(BUN), creatinine, chloride, sodium, potassium, glucose, international normalized ratio(INR), prothrombin time (PT), partial thromboplastin time(PTT), mechanical ventilation status, length of hospital stays, and length of ICU stays. If a variable was recorded multiple times within the initial 24-hour period, the mean value was utilized.The AG values were calculated using the formula: AG (mmol/L) = sodium + potassium—chloride–bicarbonate [14],while ACAG values were calculated using the formula: ACAG(tendency) = [4.4—albumin(g/dl)] * 2.5 + AG [15]. The primary outcome measure was in-hospital mortality,while the secondary outcome measures included mortality at 30-day and 90-day.

## Statistical analysis

Statistical analysis was performed by IBM SPSS software(Version 25.0), RStudio (Version 2022.07.0). A $P$-value $< 0.05$(two sided) was considered statistically significant. Variables with normal distributions were presented as the means ± SD and compared using a student T-test. The non-normally distributed variables were represented as medians and interquartile ranges (IQRs) and compared with the Mann-Whitney U-test. The counting variables were expressed as percentages and compared using the Chi-square test. According to the in-hospital survival outcome, the patients were classified into two distinct groups: the group that survived and the group that experienced mortality. We employed Random Forest method with multiple imputation to handle missing data [16]. Variables with a missing ratio exceeding 20% were excluded, and extreme values were mitigated using a 1% tail reduction approach [17]. Although the variable of lactate was missing by more than 20%, we included it due to its conventional role as a prognostic indicator for critical illness severity. Sensitivity analysis was performed using the complete data set. Receiver operating characteristic (ROC) curves for albumin (ALB), AG, and ACAG were plotted, and the area under the ROC curve was compared. The surv-cutpoint function [18] was employed to determine the optimal cut-off value of ACAG, which was subsequently utilized for stratifying patients into high and low ACAG level groups. Propensity score matching (PSM) analysis was conducted to minimize bias between these two patient groups. The patients were matched in a 2:1 ratio using nearest neighbor algorithm with a caliper width of 0.3, and standardized mean differences (SMDs) were subsequently calculated post-matching to assess balance between the groups. Cox

proportional hazards models and subgroup analysis were utilized to examine the association between ACAG levels and in-hospital mortality among trauma patients. The log-rank test and survival curve analysis were performed to compare in-hospital, 30-day and 90-day survival rates between high ACAG level group and low ACAG level group before and after PSM.

## Results

### Baseline characteristics

A total of 1038 eligible patients were ultimately included in our study, obtained from MIMIC-Ⅲ and MIMIC-Ⅳ databases (Fig 1). The patients were stratified into two cohorts based on their hospital survival outcome, comprising 900 patients in the survival cohort and 138 patients in the mortality cohort. The mortality group exhibited a significantly prolonged duration of ICU stay compared to the survival group, whereas individuals who survived had a comparatively shorter overall length of hospital stay. The mortality group exhibited higher incidences of liver disease, congestive heart failure, cancer, and diabetes in comparison to the survival group. Additionally, SOFA score, SAPSⅡ score, APSⅢ score, OASIS score, age, respiratory rate, anion gap, ACAG, sodium, BUN, creatinine, lactate, INR, PT PTT, glucose, mechanical ventilation rate, mechanical ventilation duration were all lower among survivors than non-survivors. Conversely MBP, hematocrit, hemoglobin, platelet, albumin, bicarbonate were lower in non-survivors as opposed to survivors. Furthermore racial characteristics exhibited statistical differences between both groups (Table 1).

The number of complete samples before and after the imputation of lactate value was 634 and 1038, with a median of 2.40(1.60,3.50)mmol/L and 2.30(1.55,3.30)mmol/L, respectively. Non-parametric testing revealed no significant difference between the two groups (Z = -1.461,

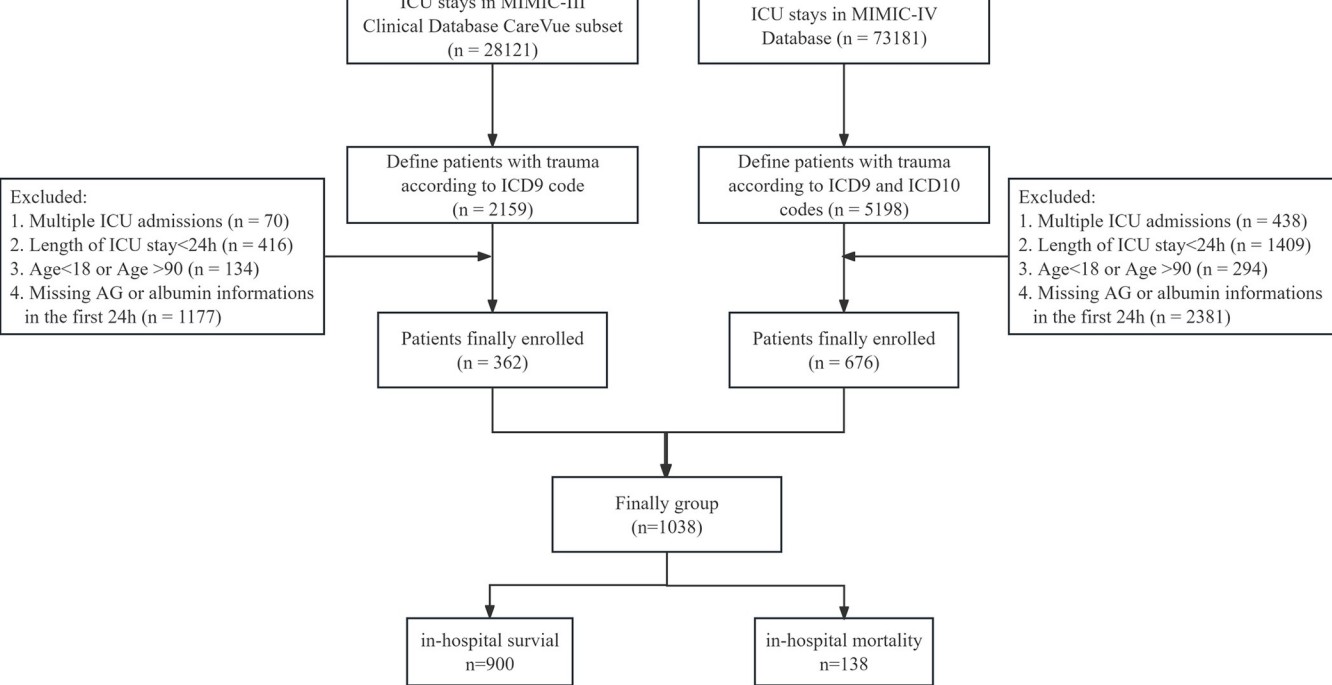

**Fig 1. Patient selection flowchart.** AG: anion gap. ICU: intensive care unit. ICD10: tenth version of the International Classifcation of Disease. ICD9: ninth version of the International Classifcation of Disease.

**Table 1. Characteristics of the study population between in-hospital survival and in-hospital mortality group.**

| Variables | Overall | in-hospital survival | in-hospital mortality | t/z/χ² | P |
|---|---|---|---|---|---|
| N | 1038 | 900 | 138 | | |
| Age (year) | 58.36 [39.19, 74.59] | 56.23 [38.11, 72.91] | 68.40 [52.66, 80.63] | -5.293 | <0.001 |
| Female, n(%) | 335 (32.3) | 280 (31.1) | 55 (39.9) | 4.185 | 0.051 |
| Weight (kg) | 76.50 [65.00, 89.10] | 76.70 [65.00, 90.00] | 74.75 [63.85, 85.00] | -0.996 | 0.319 |
| Race, n(%) | | | | 12.097 | 0.017 |
| Others | 267 (25.7) | 222 (24.7) | 45 (32.6) | | |
| White | 656 (63.2) | 570 (63.3) | 86 (62.3) | | |
| Black | 39 (3.8) | 37 (4.1) | 2 (1.4) | | |
| Hispanic | 55 (5.3) | 54 (6.0) | 1 (0.7) | | |
| Asian | 21 (2.0) | 17 (1.9) | 4 (2.9) | | |
| *Vital signs* | | | | | |
| Heart rate (bpm) | 86.07 [75.66, 100.00] | 86.07 [76.03, 99.41] | 86.42 [74.11, 102.80] | -0.269 | 0.788 |
| MBP (mmHg) | 81.08 [74.22, 88.31] | 81.43 [74.41, 88.44] | 79.40 [72.60, 86.45] | -2.118 | 0.034 |
| Respiratory rate (bpm) | 18.25 [16.37, 20.70] | 18.11 [16.32, 20.41] | 19.75 [17.73, 22.30] | -4.492 | <0.001 |
| Temperature (˚C) | 37.06 [36.73, 37.47] | 37.07 [36.74, 37.46] | 36.99 [36.52, 37.62] | -0.952 | 0.341 |
| SpO2 (%) | 98.14 [96.79, 99.30] | 98.08 [96.80, 99.27] | 98.61 [96.78, 99.47] | -1.869 | 0.062 |
| *Scoring systems* | | | | | |
| GCS | 15.00 [13.00, 15.00] | 15.00 [13.00, 15.00] | 15.00 [12.00, 15.00] | -0.693 | 0.488 |
| SOFA | 4.00 [2.00, 6.00] | 3.00 [2.00, 5.00] | 6.00 [4.00, 9.00] | -8.152 | <0.001 |
| SAPSII | 31.00 [23.00, 41.00] | 29.00 [22.00, 38.00] | 45.00 [34.00, 52.75] | -10.523 | <0.001 |
| APSIII | 38.00 [28.00, 52.00] | 36.00 [28.00, 48.00] | 54.50 [41.00, 73.00] | -9.010 | <0.001 |
| OASIS | 32.00 [27.00, 38.00] | 31.00 [26.00, 37.00] | 39.00 [34.00, 43.00] | -9.825 | <0.001 |
| *Laboratory parameters* | | | | | |
| Hematocrit (%) | 33.24 [29.47, 37.37] | 33.74 [29.72, 37.79] | 30.65 [27.95, 34.64] | -4.915 | <0.001 |
| Hemoglobin (g/dl) | 11.50 [10.07, 12.89] | 11.62 [10.20, 13.00] | 10.54 [9.42, 11.70] | -5.609 | <0.001 |
| Platelet (10^9/L) | 192.67 [144.57, 241.50] | 196.00 [151.00, 244.27] | 165.50 [112.75, 218.42] | -4.744 | <0.001 |
| Wbc (10^9/L) | 11.27 [8.40, 14.44] | 11.26 [8.42, 14.25] | 11.32 [8.27, 16.72] | -1.328 | 0.184 |
| Albumin (g/dl) | 3.40 [2.90, 3.80] | 3.50 [3.00, 3.85] | 3.20 [2.63, 3.60] | -5.031 | <0.001 |
| Anion gap (mmol/L) | 14.10 [12.21, 16.00] | 14.00 [12.00, 16.00] | 16.00 [13.53, 18.00] | -5.990 | <0.001 |
| ACAG (mmol/L) | 16.50 [14.75, 18.75] | 16.38 [14.66, 18.42] | 18.80 [16.25, 21.75] | -7.600 | <0.001 |
| Bicarbonate (mmol/L) | 23.00 [20.84, 25.46] | 23.08 [21.00, 25.50] | 21.00 [19.00, 23.79] | -5.805 | <0.001 |
| Bun (mg/dL) | 14.00 [10.00, 20.15] | 13.50 [10.00, 19.33] | 18.00 [11.68, 29.16] | -5.138 | <0.001 |
| Creatinine (mg/dL) | 0.83 [0.70, 1.10] | 0.82 [0.68, 1.05] | 1.00 [0.70, 1.40] | -4.009 | <0.001 |
| Sodium (mmol/L) | 139.50 [137.00, 141.67] | 139.38 [137.00, 141.50] | 140.67 [138.36, 143.40] | -4.487 | <0.001 |
| Potassium (mmol/L) | 4.02 [3.74, 4.37] | 4.01 [3.75, 4.38] | 4.02 [3.73, 4.34] | -0.237 | 0.813 |
| Lactate(mmol/L) | 2.30 [1.55, 3.30] | 2.26 [1.50, 3.16] | 2.78 [1.87, 4.67] | -4.729 | <0.001 |
| INR | 1.20 [1.10, 1.34] | 1.17 [1.10, 1.30] | 1.31 [1.13, 1.60] | -5.954 | <0.001 |
| PT | 13.30 [12.20, 14.70] | 13.20 [12.12, 14.40] | 14.53 [12.93, 17.03] | -6.638 | <0.001 |
| PTT | 27.64 [25.20, 31.30] | 27.38 [24.94, 30.52] | 30.24 [26.64, 36.15] | -5.926 | <0.001 |
| Glucose (mg/dL) | 133.00 [112.89, 159.25] | 129.68 [111.19, 154.56] | 157.07 [133.05, 181.37] | -6.937 | <0.001 |
| *Comorbidities* | | | | | |
| Liver disease (%) | 90 (8.7) | 68 (7.6) | 22 (15.9) | 10.627 | 0.002 |
| Paraplegia (%) | 43 (4.1) | 35 (3.9) | 8 (5.8) | 1.097 | 0.413 |
| Chronic pulmory disease (%) | 132 (12.7) | 118 (13.1) | 14 (10.1) | 0.948 | 0.403 |
| Congestive heart failure (%) | 119 (11.5) | 95 (10.6) | 24 (17.4) | 5.508 | 0.028 |
| Peripheral vascular disease (%) | 32 (3.1) | 27 (3.0) | 5 (3.6) | 0.156 | 0.897 |
| Renal disease (%) | 69 (6.6) | 54 (6.0) | 15 (10.9) | 4.572 | 0.051 |

*(Continued)*

**Table 1.** (Continued)

| Variables | Overall | in-hospital survival | in-hospital mortality | t/z/$\chi^2$ | P |
|---|---|---|---|---|---|
| Cancer (%) | 382 (36.8) | 312 (34.7) | 70 (50.7) | 13.266 | <0.001 |
| Diabetes (%) | 430 (41.4) | 357 (39.7) | 73 (52.9) | 8.634 | 0.004 |
| *Treatment* | | | | | |
| Ventilation (%) | 649 (62.5) | 525 (58.3) | 124 (89.9) | 50.739 | <0.001 |
| Ventilation duration (hours) | 18.00 [0.00, 92.00] | 12.47 [0.00, 76.93] | 84.92 [38.55, 163.38] | -8.625 | <0.001 |
| Length of hospital stays (days) | 8.73 [4.86, 16.69] | 9.35 [5.37, 17.40] | 5.93 [3.35, 11.31] | -5.393 | <0.001 |
| Length of icu stays (days) | 3.24 [1.78, 7.54] | 2.94 [1.74, 7.01] | 5.51 [2.63, 8.46] | -4.352 | <0.001 |

MAP: mean arterial pressure. Wbc: white blood cell. ACAG: albumin corrected anion gap. INR: international normalized ratio. PT: prothrombin time. PTT: partial thromboplastin time. GCS: Glasgow Coma Scale. SOFA: Sequential Organ Failure Assessment. SAPS II: Simplified Acute Physiology Scores II. APS III: Acute Physiology Score III. OASIS: Oxford Acute Severity of Illness Score.

P = 0.144). The scatter plotted in S1 Fig illustrated the distribution of lactate values before and after imputation. The imputed data exhibited the same distribution as the observed data, indicating that the missingness was completely at random (MCAR). Sensitivity analyses conducted using the complete dataset, excluding cases with missing lactate values, demonstrated that the unadjusted ACAG and the results adjusted for model 1 and model 2 (S1 Table in S1 File) did not significantly differ from those obtained using imputed lactate data (Table 4). This confirmed the robustness of the imputation method and indicated that the missing components did not have an impact on the final results.

## ROC curve analysis

The predictive efficacy of SOFA, ACAG, AG, and ALB in assessing in-hospital mortality among trauma patients was compared using ROC curve analysis. The AUC (95%CI) values for SOFA, ACAG, AG, and ALB were 0.713(0.666–0.761), 0.701 (0.652–0.749), 0.658 (0.607–0.709), and 0.633 (0.583–0.682), respectively (Fig 2). ACAG demonstrated superior predictive ability over AG for in-hospital mortality (Z = -3.420, P < 0.001) as well as over ALB(Z = -2.381, P = 0.017). Furthermore, it did not demonstrate a statistically significant difference when compared to sofa (Z = 0.425, P = 0.671).

## Determination of optimal cut-off value for survival analysis

The optimal cut-off point of ACAG for predicting in-hospital mortality in trauma patients was determined to be 20.375mmol/L using the surv_cutpoint function from the R package survminer in the R programming language. The low level ACAG group was defined as having ACAG< 20.375mmol/L, while the high level ACAG group was defined as having ACAG≥20.375mmol/L. These two groups of patients exhibited the most significant disparity (Fig 3).

## Post-PSM characteristics

The covariates (excluding AG, ACAG, and outcome variables) at baseline were included in the propensity score matching (PSM) analysis. We employed the nearest neighbor algorithm with a caliper width of 0.3 and maintained a 2:1 ratio between the control group and treatment group. Following matching, the cohorts exhibited excellent balance, with more comparable ACAG observed between the two groups (Fig 4). Table 2 and Fig 5 present the characteristics

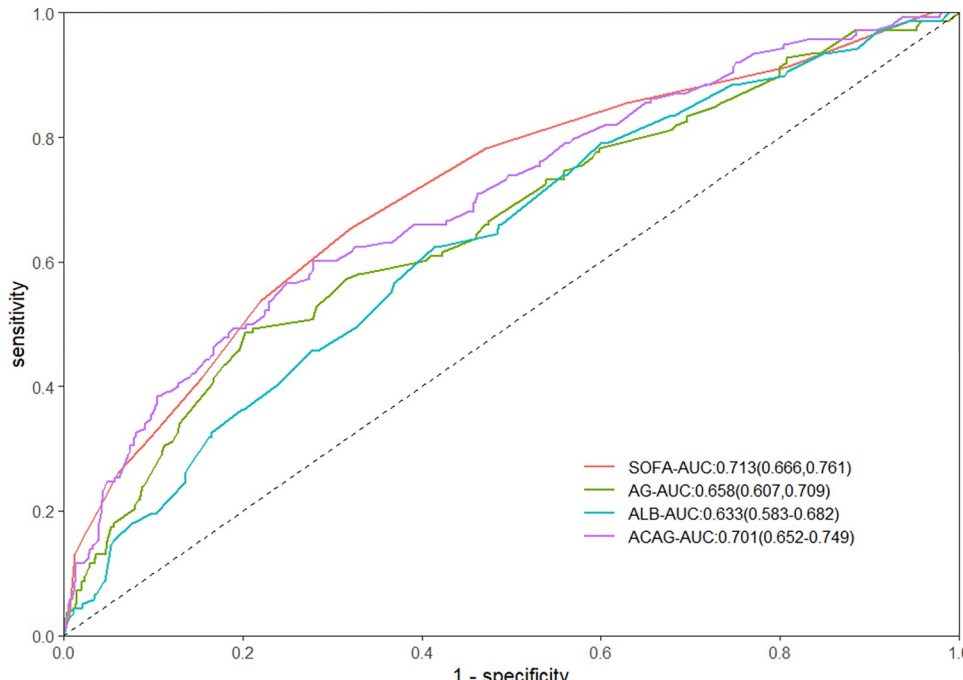

**Fig 2. Receiver-operating characteristic curves of the SOFA, ACAG, ALB and AG to predict in-hospital mortality among trauma patients.** SOFA: Sequential Organ Failure Assessment. ACAG: albumin corrected anion gap; ALB: albumin; AG: anion gap.

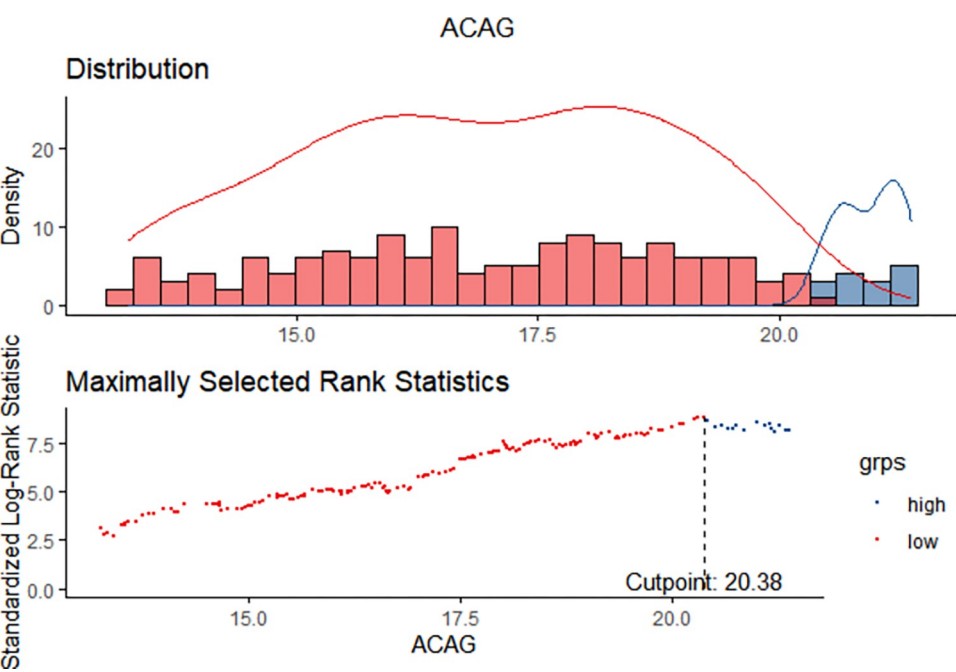

**Fig 3. The best cutoff value of ACAG was taken by the K-M curve with log-rank test.** ACAG: albumin corrected anion gap.

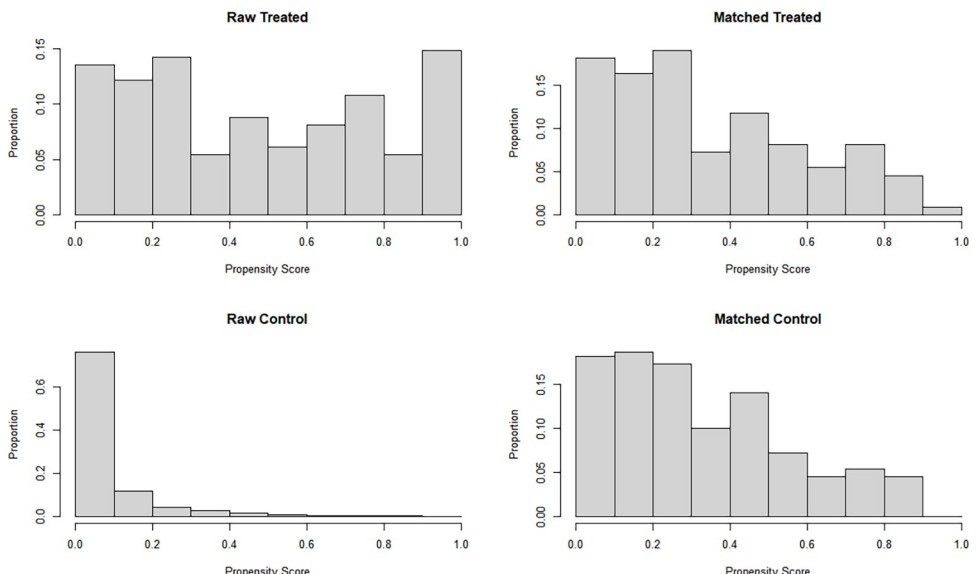

**Fig 4.** Jitter plot of distribution of propensity scores (A). Histogram of distribution of propensity scores (B).

and standardized mean differences (SMDs) of patients in both high and low ACAG groups before and after PSM. The SMDs of the matched variables were all below 0.1 after PSM.

## Subgroup analyses

The association between ACAG and in-hospital mortality was also explored using subgroup analysis (Table 3). Overall, significant interactions were not observed in most strata.

## Outcome measurement

To determine whether high ACAG level independently contribute to increased in-hospital mortality among trauma patients, we conducted univariate and multivariate Cox regression

**Table 2. Characteristics of the study population were compared between low and high ACAG groups before and after PSM.**

| Variables | Before PSM | | | After PSM | | |
|---|---|---|---|---|---|---|
| | ACAG < 20.375mmol/L | ACAG ≥ 20.375mmol/L | P | ACAG < 20.375mmol/L | ACAG < 20.375mmol/L | P |
| N | 890 | 148 | | 181 | 110 | |
| Age (year) | 57.37 [38.16, 73.92] | 62.13 [47.75, 77.26] | 0.016 | 63.61 [41.49, 78.31] | 60.22 [44.85, 75.27] | 0.696 |
| Female, n(%) | 280 (31.5) | 55 (37.2) | 0.201 | 58 (32.0) | 34 (30.9) | 0.943 |
| Weight (kg) | 76.60 [64.43, 89.83] | 75.80 [66.15, 87.62] | 0.659 | 75.65 [63.60, 90.00] | 77.85 [68.40, 88.90] | 0.804 |
| Race, n(%) | | | 0.216 | | | 0.498 |
| Others | 224 (25.2) | 43 (29.1) | | 47 (26.0) | 32 (29.1) | |
| White | 567 (63.7) | 89 (60.1) | | 112 (61.9) | 64 (58.2) | |
| Black | 34 (3.8) | 5 (3.4) | | 7 (3.9) | 5 (4.5) | |
| Hispanic | 50 (5.6) | 5 (3.4) | | 13 (7.2) | 5 (4.5) | |
| Asian | 15 (1.7) | 6 (4.1) | | 2 (1.1) | 4 (3.6) | |
| *Vital signs* | | | | | | |
| Heart rate (bpm) | 85.48 [75.23, 98.35] | 91.59 [79.39, 106.09] | 0.002 | 92.56 [82.23, 103.24] | 92.53 [79.66, 105.60] | 0.846 |
| MBP (mmHg) | 81.57 [74.68, 88.31] | 77.73 [71.71, 87.84] | 0.004 | 80.96 [74.00, 88.27] | 80.39 [72.94, 89.56] | 0.62 |
| Respiratory rate (bpm) | 18.11 [16.31, 20.38] | 19.77 [17.11, 22.50] | <0.001 | 19.30 [16.89, 21.50] | 19.52 [17.04, 22.34] | 0.365 |
| Temperature (˚C) | 37.08 [36.74, 37.48] | 37.00 [36.60, 37.33] | 0.010 | 37.07 [36.69, 37.47] | 37.02 [36.71, 37.44] | 0.924 |
| SpO2 (%) | 98.20 [96.84, 99.35] | 97.92 [96.51, 98.91] | 0.041 | 98.20 [96.83, 99.36] | 97.94 [96.70, 99.04] | 0.293 |
| *Scoring systems* | | | | | | |
| GCS | 15.00 [13.00, 15.00] | 15.00 [13.00, 15.00] | 0.604 | 15.00 [14.00, 15.00] | 15.00 [13.00, 15.00] | 0.161 |
| SOFA | 3.00 [2.00, 5.00] | 6.00 [4.00, 9.00] | <0.001 | 5.00 [3.00, 7.00] | 5.00 [3.00, 7.00] | 0.436 |
| SAPSII | 30.00 [22.00, 39.00] | 41.00 [30.00, 51.00] | <0.001 | 36.00 [28.00, 45.00] | 36.00 [28.00, 46.00] | 0.596 |
| APSIII | 37.00 [28.00, 48.00] | 55.50 [37.00, 73.25] | <0.001 | 46.00 [33.00, 62.00] | 47.00 [35.00, 62.75] | 0.509 |
| OASIS | 32.00 [27.00, 37.00] | 36.00 [31.00, 42.00] | <0.001 | 33.00 [29.00, 40.00] | 35.00 [30.00, 40.00] | 0.251 |
| *Laboratory parameters* | | | | | | |
| Hematocrit (%) | 33.51 [29.74, 37.70] | 31.26 [28.01, 35.93] | <0.001 | 31.28 [28.50, 35.83] | 31.92 [28.57, 36.86] | 0.677 |
| Hemoglobin (g/dl) | 11.60 [10.21, 13.00] | 10.65 [9.24, 12.09] | <0.001 | 10.70 [9.73, 12.27] | 10.88 [9.61, 12.32] | 0.667 |
| Platelets (10^9/L) | 195.17 [151.00, 242.83] | 166.83 [108.60, 239.75] | <0.001 | 172.83 [125.50, 234.00] | 172.50 [109.50, 240.50] | 0.747 |
| Wbc (10^9/L) | 11.25 [8.43, 14.25] | 11.57 [8.12, 15.96] | 0.207 | 11.50 [8.50, 15.07] | 11.57 [8.50, 15.28] | 0.861 |
| Albumin (g/dl) | 3.50 [3.00, 3.85] | 3.00 [2.50, 3.60] | <0.001 | 3.10 [2.60, 3.50] | 3.20 [2.56, 3.70] | 0.938 |
| Bicarbonate (mmol/L) | 23.50 [21.37, 25.50] | 20.00 [17.50, 21.67] | <0.001 | 20.67 [19.25, 22.50] | 20.33 [19.00, 22.00] | 0.206 |
| Bun (mg/dL) | 13.50 [10.00, 19.00] | 18.58 [11.33, 34.29] | <0.001 | 15.43 [11.33, 23.60] | 15.54 [10.08, 28.92] | 0.87 |
| Creatinine (mg/dL) | 0.80 [0.67, 1.02] | 1.10 [0.77, 1.71] | <0.001 | 0.90 [0.75, 1.30] | 0.94 [0.72, 1.40] | 0.664 |
| Sodium (mmol/L) | 139.50 [137.00, 141.50] | 140.00 [136.88, 142.75] | 0.092 | 140.25 [137.17, 142.33] | 140.00 [136.12, 142.00] | 0.553 |
| Potassium (mmol/L) | 4.00 [3.73, 4.30] | 4.16 [3.80, 4.62] | 0.001 | 4.14 [3.87, 4.50] | 4.05 [3.74, 4.55] | 0.522 |
| Lactate(mmol/L) | 2.20 [1.50, 3.03] | 3.81 [2.12, 5.41] | <0.001 | 3.03 [2.07, 4.20] | 3.00 [1.91, 4.54] | 0.983 |
| INR | 1.17 [1.10, 1.30] | 1.29 [1.13, 1.55] | <0.001 | 1.22 [1.10, 1.40] | 1.23 [1.10, 1.46] | 0.494 |
| PT | 13.20 [12.20, 14.42] | 14.31 [12.80, 16.65] | <0.001 | 13.70 [12.53, 15.25] | 13.68 [12.33, 15.95] | 0.793 |
| PTT | 27.40 [25.04, 30.50] | 30.31 [26.61, 35.85] | <0.001 | 28.50 [25.57, 33.20] | 29.80 [25.61, 34.52] | 0.455 |
| Glucose (mg/dL) | 132.00 [112.45, 156.50] | 142.85 [114.75, 183.70] | 0.002 | 141.67 [119.67, 166.50] | 139.67 [113.00, 175.19] | 0.516 |
| *Comorbidities* | | | | | | |
| Liver disease, n(%) | 59 (6.6) | 31 (20.9) | <0.001 | 24 (13.3) | 15 (13.6) | 1 |
| Paraplegia, n(%) | 33 (3.7) | 10 (6.8) | 0.133 | 13 (7.2) | 9 (8.2) | 0.933 |
| Chronic pulmory disease, n(%) | 118 (13.3) | 14 (9.5) | 0.250 | 14 (7.7) | 9 (8.2) | 1 |
| Congestive heart failure, n(%) | 92 (10.3) | 27 (18.2) | 0.008 | 29 (16.0) | 21 (19.1) | 0.608 |
| Peripheral vascular disease, n(%) | 29 (3.3) | 3 (2.0) | 0.585 | 3 (1.7) | 3 (2.7) | 0.844 |
| Renal disease, n(%) | 47 (5.3) | 22 (14.9) | <0.001 | 17 (9.4) | 14 (12.7) | 0.485 |
| Cancer, n(%) | 331 (37.2) | 51 (34.5) | 0.585 | 60 (33.1) | 38 (34.5) | 0.907 |

*(Continued)*

**Table 2.** (Continued)

| Variables | Before PSM | | | After PSM | | |
|---|---|---|---|---|---|---|
| | ACAG < 20.375mmol/L | ACAG ≥ 20.375mmol/L | *P* | ACAG < 20.375mmol/L | ACAG < 20.375mmol/L | *P* |
| Diabetes, n(%) | 367 (41.2) | 63 (42.6) | 0.830 | 77 (42.5) | 50 (45.5) | 0.716 |
| *Treatment* | | | | | | |
| Ventilation, n(%) | 546 (61.3) | 103 (69.6) | 0.068 | 121 (66.9) | 75 (68.2) | 0.916 |
| Ventilation duration (hours) | 15.36 [0.00, 88.04] | 35.09 [0.00, 124.94] | 0.012 | 40.00 [0.00, 150.00] | 28.38 [0.00, 124.17] | 0.574 |
| *Outcomes* | | | | | | |
| Length of hospital stays (days) | 8.77 [4.90, 16.56] | 8.18 [4.75, 16.97] | 0.820 | 11.41 [5.91, 21.36] | 8.81 [5.62, 16.58] | 0.08 |
| Length of icu stays (days) | 3.10 [1.77, 7.33] | 3.92 [1.91, 9.79] | 0.041 | 4.16 [2.01, 11.20] | 3.92 [2.00, 10.06] | 0.513 |
| in-hospital mortality, n(%) | 85 (9.6) | 53 (35.8) | <0.001 | 31 (17.1) | 35 (31.8) | 0.006 |
| 30-day mortality, n(%) | 105 (11.8) | 57 (38.5) | <0.001 | 38 (21.0) | 38 (34.5) | 0.016 |
| 90-day mortality, n(%) | 136 (15.3) | 65 (43.9) | <0.001 | 48 (26.5) | 43 (39.1) | 0.035 |

MAP: mean arterial pressure. Wbc: white blood cell. INR: international normalized ratio. PT: prothrombin Time. PTT: partial thromboplastin time. GCS: Glasgow Coma Scale. SOFA: Sequential Organ Failure Assessment. SAPS II: Simplified Acute Physiology Scores II. APS III: Acute Physiology Score III. OASIS: Oxford Acute Severity of Illness Score. PSM: propensity score matching.

analyses (Table 4). In the crude univariate models, high ACAG level was significantly associated with an increased risk of in-hospital mortality (unadjusted HR = 4.451; 95%CI:3.157–6.276). After adjusting for age, race, sex, liver disease, congestive heart failure, renal disease,

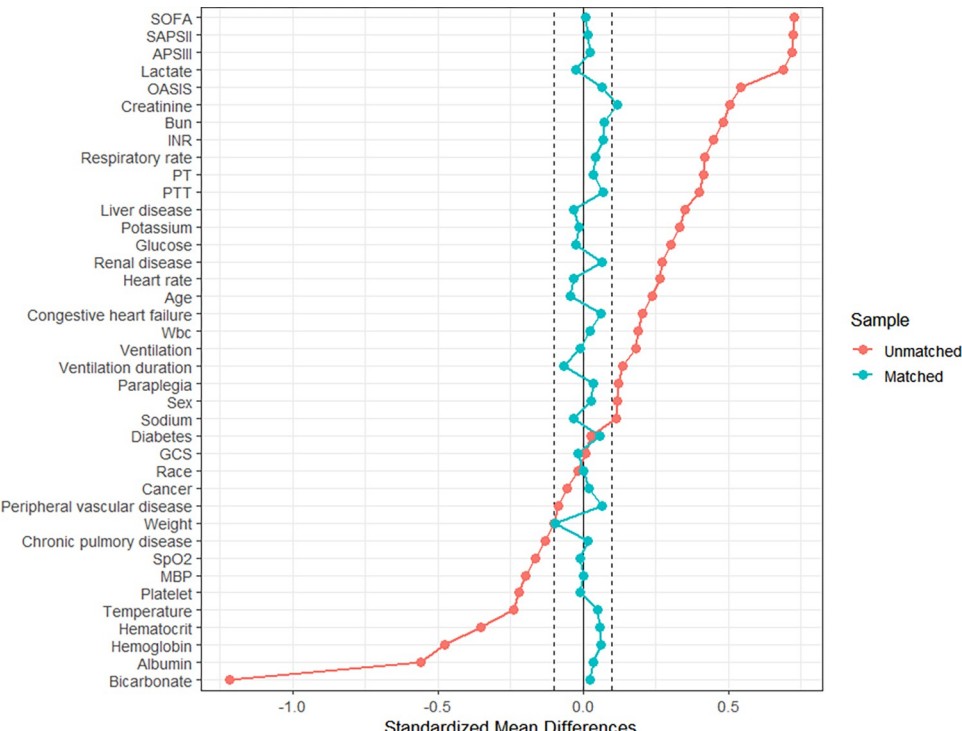

**Fig 5. Standardized mean differences (SMDs) of variables before and after matching.** MAP: mean arterial pressure. Wbc: white blood cell. ACAG: albumin corrected anion gap. INR: international normalized ratio. PT: prothrombin time. GCS: Glasgow Coma Scale. PTT: partial thromboplastin time. SOFA: Sequential Organ Failure Assessment. SAPS II: Simplified Acute Physiology Scores II. APS III: Acute Physiology Score III. OASIS: Oxford Acute Severity of Illness Score.

**Table 3. Subgroup analysis of the association between different levels of ACAG and in-hospital mortality.**

| Variable | N | Hazard ratio(95%CI) | P | P for interaction |
|---|---|---|---|---|
| Age | | | | 0.329 |
| <62 | 143 | 2.93(1.24–6.92) | 0.014 | |
| ≥62 | 148 | 1.75(0.96–3.19) | 0.067 | |
| Sex | | | | 0.386 |
| Female | 199 | 1.70(0.93–3.11) | 0.086 | |
| Male | 92 | 2.73(1.21–6.14) | 0.016 | |
| Race | | | | 0.338 |
| Others | 115 | 1.51(0.74–3.09) | 0.261 | |
| White | 176 | 2.45(1.27–4.72) | 0.008 | |
| SOFA | | | | 0.483 |
| <5 | 133 | 2.53(1.02–6.30) | 0.046 | |
| ≥5 | 158 | 1.76(0.99–3.11) | 0.054 | |
| SAPSⅡ | | | | 0.564 |
| <36 | 137 | 2.48(0.94–6.52) | 0.065 | |
| ≥36 | 154 | 1.84(1.05–3.23) | 0.032 | |
| APSⅢ | | | | 0.366 |
| <46 | 142 | 1.45(0.63–3.36) | 0.383 | |
| ≥46 | 149 | 2.37(1.30–4.30) | 0.005 | |
| OASIS | | | | 0.403 |
| <34 | 139 | 1.34(0.51–3.52) | 0.551 | |
| ≥34 | 152 | 2.16(1.23–3.80) | 0.008 | |
| Ventilation | | | | 0.338 |
| No | 95 | 1.05(0.25–4.41) | 0.942 | |
| Yes | 196 | 2.20(1.31–3.70) | 0.003 | |
| Liver disease | | | | 0.483 |
| No | 252 | 1.83(1.08–3.11) | 0.025 | |
| Yes | 39 | 3.11(0.91–10.64) | 0.071 | |
| Paraplegia | | | | 0.256 |
| No | 269 | 1.81(1.09–3.02) | 0.022 | |
| Yes | 22 | 4.76(0.92–24.63) | 0.063 | |
| Chronic pulmory disease | | | | 0.698 |
| No | 268 | 1.94(1.17–3.21) | 0.01 | |
| Yes | 23 | 2.92(0.49–17.54) | 0.242 | |
| Congestive heart failure | | | | 0.777 |
| No | 241 | 2.05(1.20–3.52) | 0.009 | |
| Yes | 50 | 1.69(0.57–5.03) | 0.345 | |
| Renal disease | | | | 0.283 |
| No | 260 | 2.19(1.30–3.70) | 0.003 | |
| Yes | 31 | 0.96(0.26–3.57) | 0.949 | |
| Cancer | | | | 0.363 |
| No | 193 | 1.60(0.78–3.24) | 0.189 | |
| Yes | 98 | 2.56(1.31–5.00) | 0.006 | |
| Diabetes | | | | 0.446 |
| No | 164 | 2.41(1.17–4.96) | 0.017 | |
| Yes | 127 | 1.67(0.87–3.21) | 0.123 | |

**Table 4. Cox proportional hazard analysis of ACAG of in-hospital mortality in patients with trauma.**

| Variable | Crude | | Model 1 | | Model 2 | | Model3 | |
|---|---|---|---|---|---|---|---|---|
| | HR(95% CI) | *P* | HR(95% CI) | *P* | HR(95% CI) | *P* | HR(95% CI) | *P* |
| ACAG<20.357mmol/L | 1(ref) | | 1(ref) | | 1(ref) | | 1(ref) | |
| ACAG≥20.357mmol/L | 4.451(3.157–6.276) | <0.001 | 4.166(2.889–6.005) | <0.001 | 3.128(1.615–6.059) | 0.001 | 1.981(1.222–3.213) | 0.006 |
| Continuous | 1.212(1.163–1.262) | <0.001 | 1.177(1.126–1.231) | <0.001 | 1.111(1.027–1.203) | 0.009 | 1.121(1.040–1.209) | 0.003 |

Crude: No covariates were adjusted before PSM. Model1: adjusted for age, race, sex, liver disease, congestive heart failure, renal disease, cancer, diabetes before PSM. Model2: adjusted for age, race, sex, MBP, respiratory rate, SpO2, SOFA, SAPSII, APSIII, OASIS, hematocrit, hemoglobin, platelets, albumin, anion gap, bicarbonate, bun, creatinine, sodium, lactate, INR, PT, PTT, glucose, liver disease, congestive heart failure, renal disease, cancer, diabetes, ventilation, ventilation duration before PSM. Model3: Univariate analysis after PSM. ACAG:albumin corrected anion gap. PSM: propensity score matching.

cancer, and diabetes in Model 1 analysis, high ACAG level remained significantly associated with higher in-hospital mortality (adjusted HR = 4.166; 95%CI:2.889–6.005). Furthermore, Model 2 incorporated additional adjustments for confounding laboratory parameters and treatments. Remarkably, elevated ACAG level still independently predicted a higher risk of in-hospital mortality (adjusted HR = 3.128; 95%CI:1.615–6.059). Notably, even within the PSM matched cohort analysis, the association between high ACAG level and in-hospital mortality remained significant (HR = 1.981;95%CI:1.222–3.213). After being included as a continuous variable in the COX regression analysis, ACAG remained statistically significant in predicting in-hospital mortality. Following adjustment for various confounding variables, the hazard ratio (HR) was 1.111 (95%CI: 1.027–1.203), and within the propensity score matching (PSM) cohort, the HR was 1.121 (95%CI: 1.040–1.209).

In addition, we employed the Log-rank test to construct Kaplan-Meier survival curves for assessing the prognostic value of ACAG. Within the original cohort, there was a significantly higher mortality rate in the high ACAG group compared to the low ACAG group at in-hospital, 30-day and 90-day time points (*P*<0.001). Furthermore, within the matched cohort, no significant differences were observed in baseline between the high and low ACAG level groups. However, high level of ACAG remained statistically associated with increased risks of in-hospital mortality(P = 0.004) as well as mortality rates at 30-day (*P = 0.012*) and 90-day (*P = 0.021*) (Fig 6).

To reduce the overestimation effect of ACAG, we attempted to hierarchically split the original dataset into a training set (n = 726) and a testing set (n = 312) at a 7:3 ratio. After conducting balance testing (S2 Table in S1 File), there were no significant differences in patient characteristics between the two datasets. ACAG was included as both a continuous variable and a categorical variable in model1 and model2 of the multivariate Cox regression analysis, respectively, in the testing set. Similar and significant results were obtained as those in the training set (S3 Table in S1 File).

## Discussion

The present study conducted a retrospective analysis on clinical data from the MIMIC-III and MIMIC-IV databases to assess the potential of AG and ACAG in predicting the prognosis of trauma patients. The findings demonstrated a significant positive correlation between AG and ACAG within the initial 24-hour period of admission to ICU and the risk of in-hospital mortality among trauma patients. Hemorrhage resulting from trauma often leaded to shock and might subsequently be accompanied by dilutive coagulopathy and hypothermia, frequently associated with severe metabolic acidosis. This condition prolonged hospitalization duration and increases mortality rates [4]. The occurrence of metabolic acidosis in severe trauma

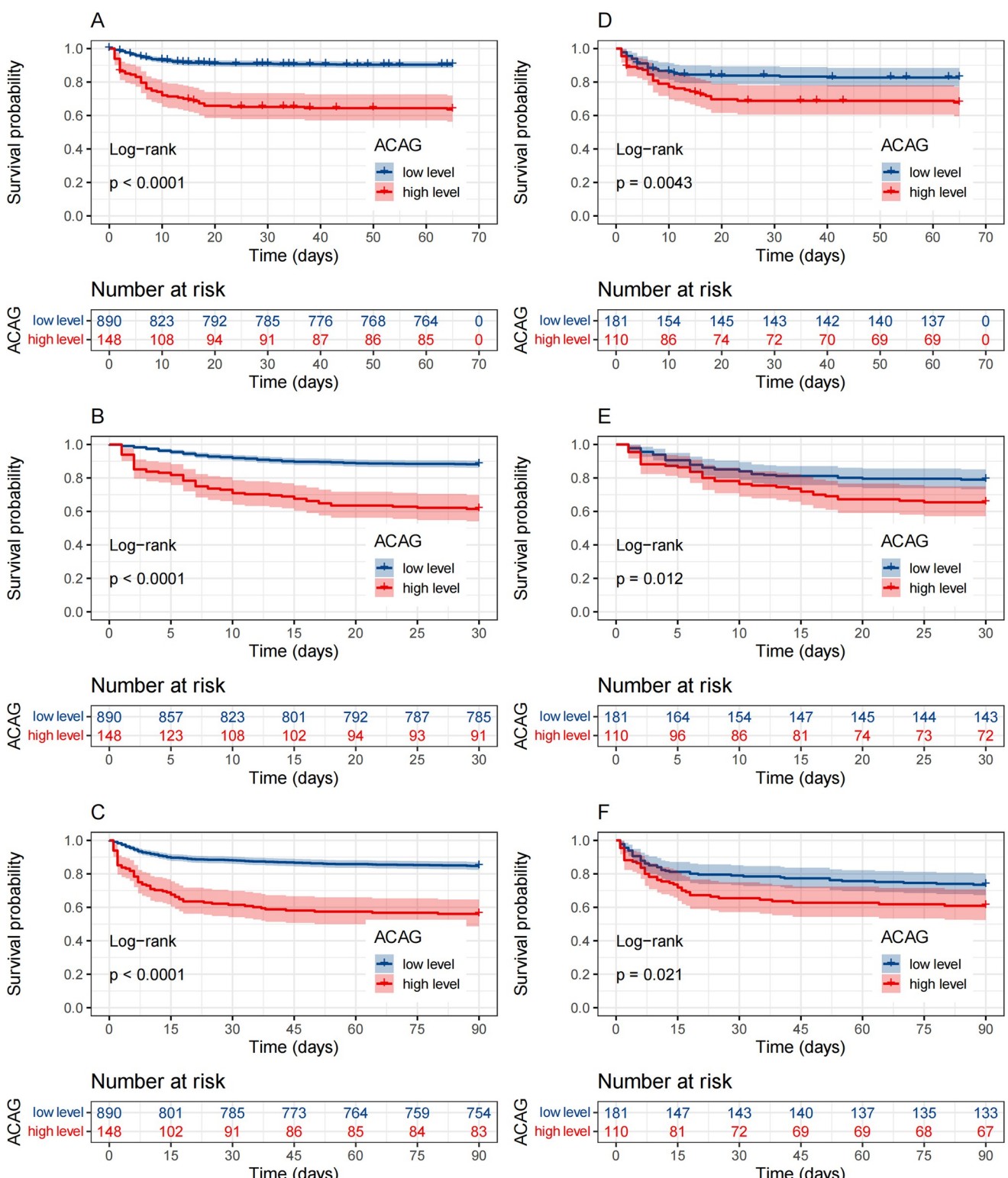

**Fig 6.** Kaplan—Meier survival curve of post-trauma patients with low level group of ACAG (blue curve. ACAG<20.375mmol/L) and high level group of ACAG (red curve. ACAG≥20.375mmol/L) at in-hospital(A, D), 30-day(B, E), 90-day(C, F) follow-up. (A-C)Reflect the results before PSM. (D-F)Reflect the results after PSM. ACAG: albumin corrected anion gap. PSM: propensity score matching.

patients could be attributed to the heightened production of organic acids, where unmeasured anions served as indicators for dissociated organic acids and were the primary contributors to metabolic acidosis [19]. As early as 1983, Stewart proposed identifying unmeasured ions through physicochemical acid-base analysis, elucidating that the charge difference between ions formed the foundation of acid-base physiology. By adhering to principles of electric neutrality and mass conservation laws, the missing charge in plasma was identified as a "gap" [20]. Both strong ion gap (SIG) and anion gap (AG) could typically serve as indicators for estimating ion gaps [21]. Kaplan et al. discovered that SIG and AG could differentiate between survivors and non-survivors of severe vascular injuries more effectively than lactic acid levels, standard base excess (SBE), or injury severity scores (ISS). The clinical utility could be enhanced by employing AG due to the relatively complex nature of SIG calculation [22]. Its easy accessibility had garnered scholarly attention in recent years, providing valuable insights into the diagnosis or prognosis of trauma patients. Leskovan et al., through a retrospective study, demonstrated that an AG level exceeding 16mmol/L was significantly associated with unfavorable clinical outcomes in elderly trauma patients [23]. Zhang et al. showed that patients with critical hip fracture and AG>12.5mmol/L had a 1.7-fold higher 30-day mortality rate compared to those with AG≤12.5mmol/L [24]. Trauma-related morbidity and mortality are frequently associated with hemorrhage, shock, tissue hypoperfusion resulting in metabolic acidosis and microcirculatory dysfunction, which could further lead to complications such as acute kidney injury (AKI), acute traumatic coagulopathy (ATC), adult respiratory distress syndrome (ARDS), ultimately culminating in fatality [21]. The two groups exhibited statistically significant differences in terms of renal function markers (creatinine, BUN), coagulation function markers (INR, PT, PTT), and others ($P<0.001$).

However, the AG could be influenced by various factors, including charged serum albumin [25]. Researches had demonstrated that for every 10 mg/L decrease in serum albumin, there was a corresponding 2.5 mmol/L decrease in AG. Hypoalbuminemia could result in a reduced measured AG, thereby concealing the presence of a high AG [26]. Adjusted corrected anion gap (ACAG) was calculated after accounting for serum albumin to mitigate this issue [27]. Hypoalbuminemia had been demonstrated to be strongly associated with unfavorable outcomes in surgical trauma patients. This correlation might arise from protein-energy malnutrition (PEM) induced by hypoalbuminemia, which could hinder wound healing, increase susceptibility to infection, exacerbate multiple organ dysfunction, and prolong hospitalization duration. Moreover, there was an elevated risk of in-hospital mortality [28]. Gonzalez et al. revealed that trauma patients with low albumin levels were more susceptible to developing traumatic endotheliopathy and subsequently experiencing protein extravasation, leading to a grim prognosis [29]. The clinical significance of ACAG lied in its ability to accurately reflect the dual pathological conditions of hypoalbuminemia and metabolic acidosis. When assessing AG levels in ICU patients, it was crucial to consider serum albumin correction as extensively as possible. In a case-control study involving 2160 individuals with acute myocardial infarction (AMI), Jian et al. found that ACAG exhibited superior predictive value compared to AG for 30-day all-cause mortality among patients in ICU. Additionally, a high ACAG level (ACAG≥21.75mmol/L) was identified as independent prognostic risk factors [30]. Moreover, numerous studies had demonstrated a significant association between ACAG and in-hospital mortality among patients suffering from cardiac arrest (CA), sepsis, and acute pancreatitis (AP) [31–33]. However, there was limited research on the correlation between ACAG and the prognosis of trauma patients. A total of 1038 trauma patients were stratified into two cohorts based on their in-hospital mortality status. Baseline data revealed a notable disparity in serum albumin within 24 hours after admission ($P < 0.001$), which could potentially impact the actual anion gap measurement. The ACAG was calculated using AG and serum albumin

values according to the calibration formula, resulting in a higher area under the curve (AUC) for ACAG compared to AG ($\triangle$AUC = 0.043, Z = -3.420, $P < 0.001$).

To further investigate the prognostic significance of ACAG in trauma patients, survival analysis was employed to demonstrate the correlation between elevated ACAG level mortality rates, including in-hospital mortality as well as 30-day, 90-day mortality rates. This study collected comprehensive and complete baseline data including demographic characteristics, vital signs, laboratory parameters, comorbidities, etc. Statistical techniques such as PSM (Propensity Score Matching) and COX regression were utilized to adjust the data, thereby enhancing the reliability of the findings. Given the absence of previous literature determining an optimal cut-off value for ACAG in trauma prognosis, we employed R language's surv-cutpoint function to identify a threshold value of 20.375mmol/L for predicting in-hospital death as the primary outcome. Subsequently, patients were categorized into two groups according to this criterion. The results demonstrated that high ACAG level remained an independent risk factor for in-hospital mortality among patients with trauma, irrespective of adjustments made through COX multivariate analysis or after PSM. Furthermore, the Kaplan-Meier survival curves provided additional evidence by confirming significantly higher rates of in-hospital, 30-day and 90-day mortality among patients with high ACAG level prior to PSM compared to those with low ACAG level. Even after achieving balance between the baseline characteristics of deceased and surviving groups following PSM, a statistically significant difference persisted between patients with high and low ACAG levels regarding various mortality outcomes. Some studies had suggested the need for lactate correction when measuring anion gap [34]. In our dataset, lactate values were missing in more than 20% of cases. However, considering the significance of lactate as a biomarker, we conducted analyses after imputing missing values and performing sensitivity analyses. We incorporated this into multivariate regression before and after PSM and observed that the results remained robust.

The present study also had certain limitations. Firstly, due to its retrospective nature and limited sample size, as well as the utilization of average values for calculating ACAG, it would be beneficial to include a larger sample or prospective external data for further validation. Secondly, despite the inclusion of numerous covariates to control for confounding variables, there was a possibility that unexplored factors might have influenced the results. Lastly, although this study focused on sever trauma patients admitted to the ICU, it did not investigate the specific trauma sites within the study population. Nevertheless, notwithstanding these limitations, our study holds significant importance in comprehending the association between ACAG and trauma.

In conclusion, elevated ACAG($>$20.375mmol/L) was found to be independently associated with in-hospital mortality as well as increased 30-day and 90-day mortality rates among critically injured patients. ACAG outperformed both albumin (ALB) and anion gap (AG) in predicting in-hospital mortality for trauma patients admitted to ICU. Given its low cost and ease of measurement, ACAG may prove useful for initial risk stratification of trauma patients, identification of high-risk individuals, and guiding clinical management.

## Supporting information

**S1 Data.**
(ZIP)

**S1 Fig.**
(TIF)

**S1 File.**
(DOCX)

## Acknowledgments

The authors express their sincere gratitude to all the participants involved in this study.

## Author Contributions

**Conceptualization:** Fei Yin.

**Data curation:** Fei Yin, Rongfei Jin.

**Formal analysis:** Fei Yin, Qiang Shi.

**Methodology:** Fei Yin, Zhenguo Qiao.

**Project administration:** Fei Yin, Yuzhou Xu.

**Resources:** Fei Yin, Yuzhou Xu.

**Software:** Fei Yin, Xiaofei Wu.

**Writing – review & editing:** Fei Yin, Zhenguo Qiao, Xiaofei Wu, Qiang Shi, Rongfei Jin, Yuzhou Xu.

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
