## [Decision Letter · Decision Letter 0]

19 Dec 2023

PONE-D-23-37440Association between albumin-corrected anion gap and in-hospital mortality of intensive care patients with trauma: a retrospective study based on MIMIC-III and IV databasesPLOS ONE

Dear Dr. Xu,

Thank you for submitting your manuscript to PLOS ONE. After careful consideration, we feel that it has merit but does not fully meet PLOS ONE’s publication criteria as it currently stands. Therefore, we invite you to submit a revised version of the manuscript that addresses the points raised during the review process.

Based on the reviewers' suggestions, the paper needs major revision.  The reviewers' comments can be found below.

We look forward to receiving your revised manuscript.

Kind regards,

Tanja Grubić Kezele, Ph.D., M.D.

Academic Editor

PLOS ONE

Journal Requirements:

Reviewers' comments:

Reviewer's Responses to Questions

**Comments to the Author**

1. Is the manuscript technically sound, and do the data support the conclusions?

Reviewer #1: No

Reviewer #2: Partly

2. Has the statistical analysis been performed appropriately and rigorously? 

Reviewer #1: No

Reviewer #2: Yes

3. Have the authors made all data underlying the findings in their manuscript fully available?

Reviewer #1: No

Reviewer #2: Yes

4. Is the manuscript presented in an intelligible fashion and written in standard English?

Reviewer #1: Yes

Reviewer #2: Yes

5. Review Comments to the Author

Reviewer #1: This article is about association between ACAG and in-hospital mortality of intensive patients with trauma using MIMIC-III and -IV databases. Though well prepared, neither the theme nor the biomarker investigated is of interest to the general audience. Here are several limitations I would like to point out so that the authors may further revise the manuscript for other journals:

1. Unlike what the authors have stated, the patients in MIMIC-III actually intersected with those in MIMIC-IV, as patients admitted to ICU from 2001-2008 (MIMIC-III) can be readmitted later in 2008-2012 (MIMIC-IV). Since identifiers of both datasets are different, the authors might not remove duplicate patients nor remove the wrong patients.

2. The association of ACAG might be overestimated since the cut-off point was chosen using statistics instead of pre-determined criteria. The authors might separate the patients into 2 groups in 8:2 ratio, respectively named as training set and validation set, so as to alleviate the possible overestimation effect.

3. Special attention should be paid to the patient group specified. Since trauma can range greatly from superficial injury to traumatic amputation. It would be impossible to figure out the appropriate spectrum to which the conclusion can be generalized. Meanwhile, the conclusion itself seemed rather dubious at the same time.

4. The manuscript lacks detailed and thorough discussion in the context of the previous literature. Rationale for this research and this biomarker should be specified. Otherwise, it seemed to be a part of huge data-mining program of limited use to clinical practice.

Reviewer #2: Thank you for the opportunity to review your research. My comments are below:

1) It is not clear when the albumin was drawn relative to the AG. For example, given that albumin is a negative acute phase reactant, if the AG was drawn on admission and the albumin was drawn close to 24 hours later, this could potentially affect the ACAG calculation. If this data on timing is not available, it should be noted in the limitations section of the discussion.

2) In Results/Baseline Characteristics, it states “…SOFA score, SAPS II score…were all higher among survivors”. I believe it should state higher among non-survivors.

3) The ROC curve analysis shows the predictive value of ACAG, AG, and albumin. It would be more meaningful to also include the Injury Severity Score or other acute severity score to determine how much ACAG adds beyond commonly available scores used to determine severity and estimate prognosis.

4) Propensity score matching is typically used to balance treatment and non-treatment groups. Its use in this manuscript is unconventional. Can you provide any references on use of PSM in this type of pathophysiology-based approach?

5) The discussion states that “…lactate values were missing by more than 20% and were subsequently excluded.” Exclusion of the lactate severely limits any conclusion that can be drawn from this study. The association between ACAG and in-hospital mortality may be confounded by or entirely due to lactate, which will increase the ACAG. Lactate should be included and can be subtracted from the ACAG and/or adjusted for in the regression analysis. Otherwise, the findings in the manuscript may entirely be due to lactate which is a well known prognostic indicator. A sensitivity analysis can be performed to ensure the missing lactate values do not contribute to uncertainty of the findings.

6. PLOS authors have the option to publish the peer review history of their article (what does this mean?). If published, this will include your full peer review and any attached files.

Reviewer #1: No

Reviewer #2: No

---

## [Author Response · Author response to Decision Letter 0]

15 Jan 2024

Dear editor and reviewers of PLOS ONE:

Our reference: PONE-D-23-37440

Title: Association between albumin-corrected anion gap and in-hospital mortality of intensive care patients with trauma: A retrospective study based on MIMIC-Ⅲ and Ⅳ databases

By: Yuzhou Xu et al

Thank you very much for your letter and for the editors’ and reviewers’ comments concerning our manuscript entitled "Association between albumin-corrected anion gap and in-hospital mortality of intensive care patients with trauma: A retrospective study based on MIMIC-Ⅲ and Ⅳ databases" (ID: PONE-D-23-37440). These comments are of great reference value to the revision and improvement of our paper and have important guiding significance to our researches. We have studied comments carefully and have made correction. We hope that the revision is acceptable and look forward to hearing from you soon. Revised portion are marked in color in the paper. The main corrections in the paper and the responds to the reviewer’s comments are as flowing:

Reviewer 1

1. Unlike what the authors have stated, the patients in MIMIC-III actually intersected with those in MIMIC-IV, as patients admitted to ICU from 2001-2008 (MIMIC-III) can be readmitted later in 2008-2012 (MIMIC-IV). Since identifiers of both datasets are different, the authors might not remove duplicate patients nor remove the wrong patients.

Since the release of MIMIC-IV, both the MIMIC-III and MIMIC-IV databases contain admissions data from 2008 to 2012, creating an overlap that complicates database analysis. To address this issue, we utilized the official MIMIC-III Clinical Database CareVue subset exclusively for our study. This subset was carefully selected to include only subject_ids not present in MIMIC-IV, ensuring that it comprises patients who were not represented in MIMIC-IV. Consequently, these two databases could be considered independent of each other without any overlapping data, thereby facilitating their combined utilization. The detailed methodology could be found on the official website: "https://physionet.org/content/mimic3-carevue/1.4/".

2. The association of ACAG might be overestimated since the cut-off point was chosen using statistics instead of pre-determined criteria. The authors might separate the patients into 2 groups in 8:2 ratio, respectively named as training set and validation set, so as to alleviate the possible overestimation effect..

Before our analysis, there was no relevant literature documenting the association between ACAG and severe trauma, and no rationale existed for this cut-off point. Therefore, we calculated it as 20.375mmol/L based on the statistical findings of our dataset. However, it was worth noting that this result closely aligned with the ACAG cut-off value reported in prognostic studies of other diseases. For instance, Jian et al., while investigating ACAG levels and 30-day all-cause mortality in acute myocardial infarction patients, determined a calculated ACAG cut-off value of 21.75mmol/L[1]. Similarly, Hu et al. demonstrated that an elevation in ACAG (≥20mmol/L) served as an independent risk factor for in-hospital mortality among cardiac arrest patients [2].

We attempted to hierarchically split the original dataset into a training set (n=726) and a testing set (n=312) at a 7:3 ratio. After conducting balance testing (Table S2), there were no significant differences in patient characteristics between the two datasets. ACAG was included as both a continuous variable and a categorical variable in model1 and model2 of the multivariate Cox regression analysis, respectively, in the testing set. Similar and significant results were obtained as those in the training set (Table S3). To further validate whether the statistical power of ACAG was overestimated, we subsequently performed propensity score matching (PSM) analysis using Table 2, Figure 4, and Figure 5. PSM was commonly used in cohort studies to mitigate confounding bias when randomization was not feasible, serving as an alternative method for multiple regression analysis with small to medium-sized sample sizes[3-5]. Through PSM, we established a new cohort where ACAG emerged as an important predictor of in-hospital mortality after adjusting for other confounding factors' balance (Table 4). After stratifying patients into two groups based on a cut-off point of 20.375mmol/L, the distinction between survival and non-survival outcomes among severe trauma patients was evident in hospital as well as at 30 and 90 days post-admission (Figure 6).

3. Special attention should be paid to the patient group specified. Since trauma can range greatly from superficial injury to traumatic amputation. It would be impossible to figure out the appropriate spectrum to which the conclusion can be generalized. Meanwhile, the conclusion itself seemed rather dubious at the same time.

We conducted a screening of trauma patients in the MIMIC database to identify those who required admission to an intensive care unit and based on a top-ranked diagnosis that matched the ICD-9 or ICD-10 trauma diagnostic codes. These criteria were employed to select critically ill patients in need of intensive care due to traumatic reasons. As Tsiklidis et al. The MIMIC III database was utilized to develop a risk prediction model for trauma patients, with the screening of trauma patients being conducted using ICD-9 codes[6].After reviewing the clinical data of 711 trauma patients, Leskovan et al. demonstrated a significant association between elevated AG without albumin correction and increased mortality and ISS scores in trauma patients. This association was also observed in patients with lower injury severity[7]. In their study, Hu et al. included all sepsis patients with a SOFA score ≥2 and suspected infection, and found that ACAG had a high predictive value for in-hospital mortality in sepsis patients[8]. Gundoğan reviewed the clinical records of pediatric patients admitted to PICU within 3 years and proposed that ACAG was an independent risk factor for death (OR = 1.064,95% CI:1.010-1.122)[9]. All these findings suggested that ACAG was an important prognostic indicator for critically ill ICU patients. The absence of AIS codes and ISS scores in this database poses challenges for conducting subgroup analyses on causes of trauma. This also aligns with our future research direction, which aims to further investigate the relationship between ACAG and different subgroups of trauma patients based on site and severity.

4. The manuscript lacks detailed and thorough discussion in the context of the previous literature. Rationale for this research and this biomarker should be specified. Otherwise, it seemed to be a part of huge data-mining program of limited use to clinical practice.

Hemorrhage resulting from trauma often leaded to shock and might subsequently be accompanied by dilutive coagulopathy and hypothermia, frequently associated with severe metabolic acidosis. This condition prolonged hospitalization duration and increases mortality rates[10]. The occurrence of metabolic acidosis in severe trauma patients could be attributed to the heightened production of organic acids, where unmeasured anions served as indicators for dissociated organic acids and were the primary contributors to metabolic acidosis[11-12]. As early as 1983, Stewart proposed identifying unmeasured ions through physicochemical acid-base analysis, elucidating that the charge difference between ions formed the foundation of acid-base physiology. By adhering to principles of electric neutrality and mass conservation laws, the missing charge in plasma was identified as a "gap"[13]. Both strong ion gap (SIG) and anion gap (AG) could typically serve as indicators for estimating ion gaps[14]. Kaplan et al. discovered that SIG and AG could differentiate between survivors and non-survivors of severe vascular injuries more effectively than lactic acid levels, standard base excess (SBE), or injury severity scores (ISS). The clinical utility could be enhanced by employing AG due to the relatively complex nature of SIG calculation[15].

References

[1] Jian L, Zhang Z, Zhou Q, Duan X, Xu H, Ge L. Association between albumin corrected anion gap and 30-day all-cause mortality of critically ill patients with acute myocardial infarction: a retrospective analysis based on the MIMIC-Ⅳ database. BMC Cardiovasc Disord. 2023;23(1):211-220. doi:10.1186/s12872-023-03200-3

[2] Hu B, Zhong L, Yuan M, Min J, Ye L, Lu J, et al. Elevated albumin corrected anion gap is associated with poor in-hospital prognosis in patients with cardiac arrest: A retrospective study based on MIMIC-Ⅳ database. Front Cardiovasc Med. 2023;10:1099003-1099010. doi:10.3389/fcvm.2023.1099003

[3] Jupiter DC. Propensity Score Matching: Retrospective Randomization?. J Foot Ankle Surg. 2017;56(2):417-420. doi:10.1053/j.jfas.2017.01.013

[4] Kane LT, Fang T, Galetta MS, et al. Propensity Score Matching: A Statistical Method. Clin Spine Surg. 2020;33(3):120-122. doi:10.1097/BSD.0000000000000932

[5] Rassen JA, Shelat AA, Myers J, Glynn RJ, Rothman KJ, Schneeweiss S. One-to-many propensity score matching in cohort studies. Pharmacoepidemiol Drug Saf. 2012;21(2):69-80. doi:10.1002/pds.3263

[6] Tsiklidis EJ, Sinno T, Diamond SL. Predicting risk for trauma patients using static and dynamic information from the MIMIC III database. PLoS One. 2022 19;17(1):e0262523. doi: 10.1371/journal.pone.0262523.

[7] Leskovan JJ, Justiniano CF, Bach JA, et al. Anion gap as a predictor of trauma outcomes in the older trauma population: correlations with injury severity and mortality. Am Surg. 2013;79(11):1203-1206.

[8] Hu T, Zhang Z, Jiang Y. Albumin corrected anion gap for predicting in-hospital mortality among intensive care patients with sepsis: A retrospective propensity score matching analysis. Clin Chim Acta. 2021;521:272-277. doi:10.1016/j.cca.2021.07.021

[9] Gündoğan Uzunay B, Köker A, Ülgen Tekerek N, Dönmez L, Dursun O. Role of Albumin-corrected Anion Gap and Lactate Clearance in Predicting Mortality in Pediatric Intensive Care Patients. Balkan Med J. 2023;40(6):430-434. doi:10.4274/balkanmedj.galenos.2023.2023-7-87

[10] Shane AI, Robert W, Arthur K, Patson M, Moses G. Acid-base disorders as predictors of early outcomes in major trauma in a resource limited setting: An observational prospective study. Pan Afr Med J. 2014;17:2. doi:10.11604/pamj.2014.17.2.2007

[11] Martin M, Murray J, Berne T, Demetriades D, Belzberg H. Diagnosis of acid-base derangements and mortality prediction in the trauma intensive care unit: the physiochemical approach [published correction appears in J Trauma. 2005 Oct;59(4):1035]. J Trauma. 2005;58(2):238-243. doi:10.1097/01.ta.0000152535.97968.4e

[12] Zingg T, Bhattacharya B, Maerz LL. Metabolic acidosis and the role of unmeasured anions in critical illness and injury. J Surg Res. 2018;224:5-17. doi:10.1016/j.jss.2017.11.013

[13] Stewart PA. Modern quantitative acid-base chemistry. Can J Physiol Pharmacol. 1983;61(12):1444-1461. doi:10.1139/y83-207

[14] Zingg T, Bhattacharya B, Maerz LL. Metabolic acidosis and the role of unmeasured anions in critical illness and injury. J Surg Res. 2018;224:5-17. doi:10.1016/j.jss.2017.11.013

[15] Kaplan LJ, Kellum JA. Initial pH, base deficit, lactate, anion gap, strong ion difference, and strong ion gap predict outcome from major vascular injury. Crit Care Med. 2004;32(5):1120-1124. doi:10.1097/01.ccm.0000125517.28517.74

Reviewer 2

1. It is not clear when the albumin was drawn relative to the AG. For example, given that albumin is a negative acute phase reactant, if the AG was drawn on admission and the albumin was drawn close to 24 hours later, this could potentially affect the ACAG calculation. If this data on timing is not available, it should be noted in the limitations section of the discussion.

Albumin values within 24 hours of admission to the ICU were not measured for every patient in MIMIC database. We observed over 3000 samples with missing albumin data, and the frequency of albumin measurement within 24 hours for samples without missing data was significantly lower compared to that of anion gap. Moreover, there was a smaller number of samples where both anion gap and albumin were measured simultaneously. These factors presented challenges in obtaining real-time ACAG from the database. Similar to other laboratory data, ACAG was calculated using the average value within 24 hours after ICU admission. This limitation of our study should be acknowledged, and it was anticipated that future research endeavors will address this issue through a real-world prospective study or validation with a larger sample size to further substantiate the relevance of real-time ACAG in determining patient outcomes following severe trauma.

2. In Results/Baseline Characteristics, it states “…SOFA score, SAPS II score…were all higher among survivors”. I believe it should state higher among non-survivors.

The errors that occurred in the writing of the results due to our negligence have been rectified in the original text, for which we sincerely apologize. SOFA score, SAPSⅡ score, APSⅢ score, OASIS score, age, respiratory rate, anion gap, ACAG, sodium, BUN, creatinine, lactate, INR, PT PTT, glucose, mechanical ventilation rate, mechanical ventilation duration were all lower among survivors than non-survivors.

3. The ROC curve analysis shows the predictive value of ACAG, AG, and albumin. It would be more meaningful to also include the Injury Severity Score or other acute severity score to determine how much ACAG adds beyond commonly available scores used to determine severity and estimate prognosis.

Thank you for your insightful feedback. We had incorporated the SOFA score into the ROC curves(Figure 2) and determined an AUC(95%CI) of 0.713(0.666-0.761) for this score. There was no statistically significant difference in AUC between ACAG and SOFA score(Z = 0.425, P = 0.671).

Figure 2 Receiver-operating characteristic curves of the SOFA, ACAG ,ALB and AG to predict in-hospital mortality among trauma patients. SOFA: Sequential Organ Failure Assessment. ACAG: albumin corrected anion gap; ALB: albumin; AG: anion gap.

4. Propensity score matching is typically used to balance treatment and non-treatment groups. Its use in this manuscript is unconventional. Can you provide any references on use of PSM in this type of pathophysiology-based approach?

In small and medium-sized sample size environments, propensity score matching (PSM) can serve as an alternative to multiple regression analysis for mitigating confounding bias in non-randomized observational cohorts. Extensive literature review reveals that PSM not only achieves balance between treatment and control groups by accounting for confounding factors but also facilitates the investigation of associations between specific factors and disease prognosis.

Tang et al. employed a multivariate logistic regression model to estimate the propensity score when investigating the association between INR and all-cause mortality in patients with cardiac arrest. They performed 1:1 matching of subjects in the low INR group and high INR group, followed by re-performing Kaplan-Meier curve analysis and COX regression analysis to examine the potential impact of INR on all-cause mortality in post-cardiac arrest patients [1]. Zhang et al., in their study on the correlation between Acidemia and 30-day/90-day mortality in AMI patients, categorized all participants into Non-acidemia group and acidemia group, implemented PSM to mitigate any imbalances between these groups, subsequently employing Kaplan-Meier survival analysis to compare patient mortality rates across both groups [2]. The method of propensity score matching (PSM) was also employed by Wu et al. in their study, wherein patients were categorized into an AF group and a non-AF group. Subsequently, the primary and secondary outcomes were compared based on the matched data obtained through PSM[3].

The optimal cut-off value of ACAG was initially determined and subsequently re-grouped in our study. Propensity score matching (PSM) was employed to mitigate bias between the two patient groups. Ultimately, KM curves were obtained with adjustment for confounding factors, revealing a significant association between ACAG and both primary and secondary outcomes.

5. The discussion states that “…lactate values were missing by more than 20% and were subsequently excluded.” Exclusion of the lactate severely limits any conclusion that can be drawn from this study. The association between ACAG and in-hospital mortality may be confounded by or entirely due to lactate, which will increase the ACAG. Lactate should be included and can be subtracted from the ACAG and/or adjusted for in the regression analysis. Otherwise, the findings in the manuscript may entirely be due to lactate which is a well known prognostic indicator. A sensitivity analysis can be performed to ensure the missing lactate values do not contribute to uncertainty of the findings.

The expression of gratitude is extended to you for bringing this matter to my attention. Although lactate values were missing in over 20% of cases within our dataset, it was imperative to incorporate lactate into the multivariate regression analysis in order to mitigate bias, given its pivotal role as a biomarker. The technique of multiple imputation was employed to handle missing lactate values. 

The number of samples with lactate values before and after imputation was 634 and 1038, with a median of 2.40(1.60,3.50)mmol/L and 2.30(1.55,3.30)mmol/L, respectively. Non-parametric testing revealed no significant difference between the two groups (Z = -1.461, P = 0.144). The scatter plot in Figure S1 illustrated the distribution of lactate values before and after imputation. The imputed data exhibited the same distribution as the observed data, indicating that the missingness is completely at random (MCAR).

Sensitivity analyses conducted using the complete dataset, excluding cases with missing lactate values, demonstrated that the unadjusted ACAG and the results adjusted for model 1 and model 2 (Table S1) did not significantly differ from those obtained using imputed lactate data (Table 4). This confirmed the robustness of the imputation method and indicated that the missing components did not have an impact on the final results.

Figure S1 Multiple imputation scatter plot of lactate. The blue dots represent the observed data, while the red dots indicate the imputed data. The ordinate axis represents the lactate value, and the abscissa axis represents the number of imputations, with 0 representing the original data column.

Table 4 Cox proportional hazard analysis of ACAG of in-hospital mortality in patients with trauma, following imputation of lactate values.

Variable Crude Model 1 Model 2

 HR(95% CI) P HR(95% CI) P HR(95% CI) P

ACAG<20.357mmol/L 1(ref) 1(ref) 1(ref) 

ACAG≥20.357mmol/L 4.451(3.157-6.276) <0.001 4.166(2.889-6.005) <0.001 3.128(1.615-6.059) 0.001

Continuous 1.212(1.163-1.262) <0.001 1.177(1.126-1.231) <0.001 1.111(1.027-1.203) 0.009

Crude: No covariates were adjusted. Model1: adjusted for age, race, sex, liver disease, congestive heart failure, renal disease, cancer, diabetes. Model2: adjusted for age, race, sex, MBP, respiratory rate, SpO2, SOFA, SAPSⅡ, APSⅢ, OASIS, hematocrit, hemoglobin, platelets, albumin, anion gap, bicarbonate, bun, creatinine, sodium, lactate, INR, PT, PTT, glucose, liver disease, congestive heart failure, renal disease, cancer, diabetes, ventilation, ventilation duration. ACAG:albumin corrected anion gap.

Table S1 Cox proportional hazard analysis of ACAG in the data set excluding cases with missing lactate values 

Variable Crude Model 1 Model 2

 HR(95% CI) P HR(95% CI) P HR(95% CI) P

ACAG<20.357mmol/L 1(ref) 1(ref) 1(ref) 

ACAG≥20.357mmol/L 4.609(3.168-6.707) <0.001 4.365(2.937-6.486) <0.001 3.259(1.536-6.915) 0.002 

Continuous 1.195(1.145-1.248) <0.001 1.161(1.109-1.216) <0.001 1.146(1.049-1.252) 0.004

Crude: No covariates were adjusted. Model1: adjusted for age, race, sex, liver disease, congestive heart failure, renal disease, cancer, diabetes. Model2: adjusted for age, race, sex, MBP, respiratory rate, SpO2, SOFA, SAPSⅡ, APSⅢ, OASIS, hematocrit, hemoglobin, platelets, albumin, anion gap, bicarbonate, bun, creatinine, sodium, lactate, INR, PT, PTT, glucose, liver disease, congestive heart failure, renal disease, cancer, diabetes, ventilation, ventilation duration. ACAG:albumin corrected anion gap. 

References

[1] Tang Y, Sun J, Yu Z, et al. Association between prothrombin time-international normalized ratio and prognosis of post-cardiac arrest patients: A retrospective cohort study. Front Public Health. 2023;11:1112623. Published 2023 Jan 20. doi:10.3389/fpubh.2023.1112623

[2] Zhang T, Guan YZ, Liu H. Association of Acidemia With Short-Term Mortality of Acute Myocardial Infarction: A Retrospective Study Base on MIMIC-III Database. Clin Appl Thromb Hemost. 2020;26:1076029620950837. doi:10.1177/1076029620950837

[3] Wu CS, Chen PH, Chang SH, et al. Atrial Fibrillation Is Not an Independent Determinant of Mortality Among Critically Ill Acute Ischemic Stroke Patients: A Propensity Score-Matched Analysis From the MIMIC-IV Database. Front Neurol. 2022;12:730244. Published 2022 Jan 17. doi:10.3389/fneur.2021.730244

Once again, thank you very much for your comments and suggestions.

Yours sincerely,

Yuzhou Xu

2024-01-15

---

## [Decision Letter · Decision Letter 1]

30 Jan 2024

PONE-D-23-37440R1Association between albumin-corrected anion gap and in-hospital mortality of intensive care patients with trauma: a retrospective study based on MIMIC-III and IV databasesPLOS ONE

Dear Dr. Xu,

Thank you for submitting your manuscript to PLOS ONE. After careful consideration, we feel that it has merit but does not fully meet PLOS ONE’s publication criteria as it currently stands. Therefore, we invite you to submit a revised version of the manuscript that addresses the points raised during the review process.

Your manuscript, entitled "Association between albumin-corrected anion gap and in-hospital mortality of intensive care patients with trauma: a retrospective study based on MIMIC-III and IV databases", has been reviewed. Your efforts to revise the manuscript are appreciated. However, the peer review process continues because Reviewer 1 has a few additional comments that the author should address. Please find the reviewer's commentary below.

We look forward to receiving your revised manuscript.

Kind regards,

Tanja Grubić Kezele, Ph.D., M.D.

Academic Editor

PLOS ONE

Journal Requirements:

Reviewers' comments:

Reviewer's Responses to Questions

**Comments to the Author**

1. If the authors have adequately addressed your comments raised in a previous round of review and you feel that this manuscript is now acceptable for publication, you may indicate that here to bypass the “Comments to the Author” section, enter your conflict of interest statement in the “Confidential to Editor” section, and submit your "Accept" recommendation.

Reviewer #1: All comments have been addressed

Reviewer #2: All comments have been addressed

2. Is the manuscript technically sound, and do the data support the conclusions?

Reviewer #1: Yes

Reviewer #2: (No Response)

3. Has the statistical analysis been performed appropriately and rigorously? 

Reviewer #1: Yes

Reviewer #2: (No Response)

4. Have the authors made all data underlying the findings in their manuscript fully available?

Reviewer #1: No

Reviewer #2: (No Response)

5. Is the manuscript presented in an intelligible fashion and written in standard English?

Reviewer #1: Yes

Reviewer #2: (No Response)

6. Review Comments to the Author

Reviewer #1: I am rather glad to review the new version of the manuscript which is much more acceptable for publication. One minor problem remained that the detailed ICD code for trauma or the specific spectrum for the alleged trauma population has been given. The authors are strongly encouraged to provide detailed ICD-9/10 they used to extract the patients on the Github repo or submit to the journal as supplementary materials. Without enough illustration, the conclusions of this article can hardly be put into clinical practice.

Reviewer #2: (No Response)

7. PLOS authors have the option to publish the peer review history of their article (what does this mean?). If published, this will include your full peer review and any attached files.

Reviewer #1: No

Reviewer #2: No

---

## [Author Response · Author response to Decision Letter 1]

4 Feb 2024

Dear editor and reviewers of PLOS ONE:

Our reference: PONE-D-23-37440R1

Title: Association between albumin-corrected anion gap and in-hospital mortality of intensive care patients with trauma: A retrospective study based on MIMIC-Ⅲ and Ⅳ databases

By: Yuzhou Xu et al

Thank you very much for your letter and for the editors’ and reviewers’ comments concerning our manuscript entitled "Association between albumin-corrected anion gap and in-hospital mortality of intensive care patients with trauma: A retrospective study based on MIMIC-Ⅲ and Ⅳ databases" (ID: PONE-D-23-37440R1). These comments are of great reference value to the revision and improvement of our paper and have important guiding significance to our researches. We have studied comments carefully and have made correction. We hope that the revision is acceptable and look forward to hearing from you soon. Revised portion are marked in color in the paper. The main corrections in the paper and the responds to the reviewer’s comments are as flowing:

Reviewer 1

1. I am rather glad to review the new version of the manuscript which is much more acceptable for publication. One minor problem remained that the detailed ICD code for trauma or the specific spectrum for the alleged trauma population has been given. The authors are strongly encouraged to provide detailed ICD-9/10 they used to extract the patients on the Github repo or submit to the journal as supplementary materials. Without enough illustration, the conclusions of this article can hardly be put into clinical practice.

We appreciate your valuable feedback, and as per your suggestion, we have incorporated the coding range for trauma-related ICD-9/10 in our article. Specifically, the ICD-9 trauma codes encompass the range of 800-959, while the ICD-10 trauma codes include S00-S99, T00-T14, and T20-T32[1-3]. 

A comprehensive inventory of trauma-related ICD-9/10 codes within the MIMIC-III/IV databases is provided in the appendix. Furthermore, we have incorporated all diagnostic codes for each patient into the original dataset. The aforementioned items have been submitted to the journal as supplementary materials.. 

References

[1] Flynn-O'Brien KT, Fallat ME, Rice TB, et al. Pediatric Trauma Assessment and Management Database: Leveraging Existing Data Systems to Predict Mortality and Functional Status after Pediatric Injury. J Am Coll Surg. 2017;224(5):933-944.e5. doi:10.1016/j.jamcollsurg.2017.01.061

[2] Clark DE, Black AW, Skavdahl DH, Hallagan LD. Open-access programs for injury categorization using ICD-9 or ICD-10. Inj Epidemiol. 2018;5(1):11-18. doi:10.1186/s40621-018-0149-8

[3] Wada T, Yasunaga H, Yamana H, et al. Development and validation of a new ICD-10-based trauma mortality prediction scoring system using a Japanese national inpatient database. Inj Prev. 2017;23(4):263-267. doi:10.1136/injuryprev-2016-042106

Furthermore, we have rectified certain formatting issues in the references. All accompanying figures have been adjusted by PACE.

Once again, thank you very much for your comments and suggestions.

Yours sincerely,

Yuzhou Xu

2024-02-04

---

## [Decision Letter · Decision Letter 2]

20 Feb 2024

Association between albumin-corrected anion gap and in-hospital mortality of intensive care patients with trauma: a retrospective study based on MIMIC-III and IV databases

PONE-D-23-37440R2

Dear Dr. Xu,

We’re pleased to inform you that your manuscript has been judged scientifically suitable for publication and will be formally accepted for publication once it meets all outstanding technical requirements.

Kind regards,

Tanja Grubić Kezele, Ph.D., M.D.

Academic Editor

PLOS ONE

Additional Editor Comments (optional):

Reviewers' comments:

Reviewer's Responses to Questions

**Comments to the Author**

1. If the authors have adequately addressed your comments raised in a previous round of review and you feel that this manuscript is now acceptable for publication, you may indicate that here to bypass the “Comments to the Author” section, enter your conflict of interest statement in the “Confidential to Editor” section, and submit your "Accept" recommendation.

Reviewer #1: All comments have been addressed

2. Is the manuscript technically sound, and do the data support the conclusions?

Reviewer #1: Yes

3. Has the statistical analysis been performed appropriately and rigorously? 

Reviewer #1: Yes

4. Have the authors made all data underlying the findings in their manuscript fully available?

Reviewer #1: Yes

5. Is the manuscript presented in an intelligible fashion and written in standard English?

Reviewer #1: Yes

6. Review Comments to the Author

Reviewer #1: (No Response)

7. PLOS authors have the option to publish the peer review history of their article (what does this mean?). If published, this will include your full peer review and any attached files.

Reviewer #1: No

---

## [Editor Report · Acceptance letter]

27 Feb 2024

PONE-D-23-37440R2 

PLOS ONE

Dear Dr. Xu, 

I'm pleased to inform you that your manuscript has been deemed suitable for publication in PLOS ONE. Congratulations! Your manuscript is now being handed over to our production team.

Kind regards, 

on behalf of

Prof. dr. Tanja Grubić Kezele 

Academic Editor

PLOS ONE